# Drone Detection Using YOLOv5

**Burchan Aydin [1],*** and **Subroto Singha [2]**

1    Department of Engineering and Technology, Texas A&M University-Commerce, Commerce, TX 75428, USA
2    Department of Computer Science and Information Systems, Texas A&M University-Commerce, Commerce, TX 75428, USA
*    Correspondence: burchan.aydin@tamuc.edu; Tel.: +1-903-886-5174

**Abstract:** The rapidly increasing number of drones in the national airspace, including those for recreational and commercial applications, has raised concerns regarding misuse. Autonomous drone detection systems offer a probable solution to overcoming the issue of potential drone misuse, such as drug smuggling, violating people's privacy, etc. Detecting drones can be difficult, due to similar objects in the sky, such as airplanes and birds. In addition, automated drone detection systems need to be trained with ample amounts of data to provide high accuracy. Real-time detection is also necessary, but this requires highly configured devices such as a graphical processing unit (GPU). The present study sought to overcome these challenges by proposing a one-shot detector called You Only Look Once version 5 (YOLOv5), which can train the proposed model using pre-trained weights and data augmentation. The trained model was evaluated using mean average precision (mAP) and recall measures. The model achieved a 90.40% mAP, a 21.57% improvement over our previous model that used You Only Look Once version 4 (YOLOv4) and was tested on the same dataset.

**Keywords:** YOLOv5; autonomous drone detection; image recognition; machine learning; mAP; unmanned aerial vehicle (UAV)





## 1. Introduction

Drones are becoming increasingly popular. Most are inexpensive, flexible, and lightweight [1]. They are utilized in a variety of industries, including the military, construction, agriculture, real estate, manufacturing, photogrammetry, sports, and photography [2,3]. There were 865,505 drones registered as of 3 October 2022, with 538,172 of them being recreational [4]. Drones can take off and land autonomously, intelligently adapt to any environment, fly to great heights, and provide quick hovering ability and flexibility [5]. Increased usage of drones, on the other hand, poses a threat to public safety; for example, their capacity to carry explosives may be used to strike public locations, such as governmental and historical monuments [6]. Drones can also be used by drug smugglers and terrorists. Moreover, the increasing number of hobbyist drone pilots could result in interference with activities, such as firefighting, disaster response efforts, and so on [7]. A list of threats that drones currently pose and a discussion of how drones are being weaponized are offered in [8]. For instance, in April 2021, two police officers in Aguililla, Michoacan, Mexico were assaulted by drones *artillados* carrying explosive devices, resulting in multiple injuries [9]. Thirteen tiny drones attacked Russian soldiers in Syria, causing substantial damage [10]. Considering the possibility of drones being used as lethal weapons [11], authorities shut down the London Gatwick airport for 18 hours due to serious drone intrusion, causing 760 flights with over 120,000 people to be delayed [12].

Detecting drones may be difficult due to the presence of similar objects in the sky, such as aircrafts, birds, and so forth. The authors of [13] used a dataset made up of drones and birds. To create the dataset, they gathered drone and bird videos and extracted images using the MATLAB image processing tool. After gathering 712 photos to train the algorithms, they utilized an 80:20 train:test split to randomly choose the training and testing images.

They examined the accuracies of three different object detectors utilizing an Intel Core i5–4200M (2.5GHZ0), 2GB DDR3 L Memory, and 1TB HDD, reaching 93%, 88%, and 80% accuracy using the CNN, SVM, and KNN, respectively. The suggested technique examined included drone-like objects, i.e., birds in the dataset; however, it required 14 minutes and 28 seconds to attain 93% accuracy for just 80 epochs using the CNN methodology. As a result, their proposed approach was not feasible for real-time implementation.

Our previously proposed technique using fine-tuned YOLOv4 [14] overcame the speed, accuracy, and model overfitting issues. In that study, we collected 2395 images of birds and drones from public sources, such as Google, Kaggle, and others. We labeled the images and divided them into two categories: drones and birds. The YOLOv4 model was then trained on the Tesla K80 GPU using the Google deep learning VM. To test the detecting speed, we recorded two drone videos of our own drones at three different heights. The trained model obtained an FPS of 20.5 and 19.0. The mAP was 74.36%. In terms of speed and accuracy, YOLOv5 surpassed prior versions of YOLO [1]. In this study, we compared the performance increase using fine-tuned YOLOv5 for the same dataset used in [14] for drone detection using fine-tuned YOLOv4. YOLOv5 recently demonstrated improved performance in identifying drones. The authors of [1] presented a method for detecting drones flying in prohibited or restricted zones. Their deep learning-based technique outperformed earlier deep learning-based methodologies in terms of precision and recall.

Our key contributions to this study were the addition of a data augmentation technique to artificially overcome data scarcity difficulties, as well as the prevention of overfitting issues utilizing a random train:test split of 70:30, the fine-tuning of the original YOLOv5 based on our collected customized dataset, the testing of the model on a wide variety of backgrounds (dark, sunny), and the testing of different views of images. The model was tested on our own videos using two drones-DJI Mavic Pro, DJI Phantom; videos were taken at three common altitudes—60 ft, 40 ft, and 20 ft.

*Paper Organization*

The rest of the research study is structured as follows. Section 2 provides background for our research. Section 3 addresses the research materials and methodologies. Section 4 covers the findings of this study. Section 5 discusses the model's complexity and uncertainty. Section 6 depicts the performance improvement and gives an argumentative discussion. Section 7 brings our paper to a conclusion.

## 2. Background

In the past, various techniques, such as radar, were used to detect drones [15]. However, it is very difficult for radar to do so, due to the low levels of electromagnetic signals that drones transmit [16]. Similarly, other techniques, such as acoustic and radio frequency-based drone detection, are costly and inaccurate [17]. Recently, machine learning-based drone detectors, such as SVM and artificial neural network classifiers, have been used to detect drones, achieving better success than radar and acoustic drone detection systems [18]. The YOLO algorithm has outperformed competitor algorithms, such as the R-CNN and SSD algorithms, due to its complex feature-learning capability with fast detection [18]. In fact, the YOLO algorithm is now instrumental in object detection tasks [19]. Many computer vision tasks use YOLO due to its faster detection with high accuracy, which makes the algorithm feasible for real-time implementation [20]. One of the latest developments, YOLOv5, has greatly improved the algorithm's performance, offering a 90% improvement over YOLOv4 [21]. In the present research, we used YOLOv5 to build an automated drone detection system and compared the results against our previous system with the YOLOv4.

UAV detection systems are designed using various techniques. We have reviewed only those studies closely related to our methodology. UAV detection can be treated as an object detection problem in deep learning. Deep learning-based object detection techniques can be divided into one-stage and two-stage detection algorithms [22]. An example of a

two-stage object detection technique is R-CNN [23]; examples of one-stage object detection techniques are YOLO [24], SSD [25], etc. The authors of [26] explained the mechanism of how object detectors work in general. Two-stage detectors use candidate object techniques, while one-stage detectors employ the sliding window technique. Thus, one-stage detectors are fast and operate in real-time [27]. YOLO is easy to train, faster, more accurate than its competitors, and can immediately train an entire image. Thus, YOLO is the most frequently used and reliable object detection algorithm [28]. It first divides an image into SXS grids and assigns a class probability with bounding boxes around the object [28]. It then uses a single convolutional network to perform the entire prediction. Conversely, R-CNNs begin by generating a large number of region proposals using a selective search method. Then, from each region proposal, a CNN is utilized to extract features. Finally, the R-CNN classifies and defines bounding boxes for distinct classes [28].

The authors of [28] used YOLOv2 to detect drones and birds, and achieved precision and recall scores above 90. The authors of [27] proposed a drone detection pipeline with three different models: faster R-CNN with ResNet–101, faster R-CNN with Inceptionv2, and SSD. After 60,000 iterations, they achieved mAP values of 0.49, 0.35, and 0.15, respectively. One example of an SSD object detector is MobileNet. MobileNetV2 was used as a classifier in [29]; the authors proposed a drone detection model where the methodology consisted of a moving object detector and a drone-bird-background classifier. The researchers trained the drone-vs-bird challenge dataset on the NVIDIA GeForce GT 1030 2GB GPU with a learning rate of 0.05. At an IoU of 0.5, their highest precision, recall, and F1 scores were 0.786, 0.910, and 0.801, respectively, after testing on three videos. The authors of [30] used YOLOv3 to detect and classify drones. The authors of [30] collected different types of drone images from the internet and videos to build a dataset. Images were annotated in the YOLO format in order to train a YOLOv3 model. An NVIDIA GeForce GTX 1050 Ti GPU was used to train the dataset with chosen parameter values, such as a learning rate of 0.0001, batch size of 64, and 150 total epochs. The best mAP value was 0.74. PyTorch, an open-source machine learning programming language, was used to train and test the YOLOv3 model.

The authors of [31] used YOLOv4 to automatically detect drones in order to integrate a trained model into a CCTV camera, thus reducing the need for manual monitoring. The authors collected their dataset from public resources such as Google images, open-source websites, etc. The images were converted into the YOLO format using free and paid image annotation tools. They fine-tuned the YOLOv4 architecture by customizing filters, max batches, subdivisions, batches, etc. After training the YOLOv4 model for 1300 iterations, the researchers achieved a mAP of 0.99. Though their mAP value was very high, they trained only 53 images and did not address model overfitting, resulting in a greater improvement scope.

The authors of [1] presented an approach based on YOLOv5. They utilized a dataset of 1359 drone images obtained from Kaggle. They fine-tuned the model on a local system with an 8 GB NVIDIA RTX2070 GPU, 16 GB of RAM, and a 1.9 GHz CPU. They employed a 60:20:20 split of the dataset for training, testing, and validation. They trained the model on top of COCO pre-trained weights and obtained a precision of 94.70%, a recall of 92.50%, and a mAP of 94.1%.

## 3. Materials and Methods

In this research, we employed a recent version of the YOLO algorithm: YOLOv5 [32]. YOLOv5 is a high-performing and fast object detection algorithm that detects objects in real-time. Drones can fly at fast speeds; thus the detection speed also needs to be high. YOLOv5 has the ability to meet this requirement. The algorithm was developed using PyTorch, an open-source deep learning framework that has made training and testing easier for customized datasets and offers outstanding detection performance. YOLOv5 consists of three parts: the backbone, neck, and head [1].

The backbone is made of a CSPNet. The CSPNet reduces the model's complexity, resulting in fewer hyperparameters and FLOPS. At the same time, it resolves vanishing and

exploding gradient issues, due to the depth of the neural networks. These improvements enhance inference speed and accuracy in object detection. Inside the CSPNet, there are several convolutional layers, four CSP bottlenecks with three convolutions, and spatial pyramid pooling. The CSPNet is responsible for extracting features from an input image and using convolutions and pooling to form a feature map that combines all extracted features. Thus, the backbone plays the role of feature extractor in YOLOv5.

The middle part of YOLOv5, often called the neck, is also known as the PANet. The PANet takes all the extracted features from the backbone and saves and sends them to the deep layers in order to perform feature fusions. These feature fusions are passed to the head so that high-level features are known to the output layer for final object detection.

The head of YOLOv5 is responsible for object detection. It consists of 1x1 convolutions that predict the class of an object, with bounding boxes around the target object and a class probability score. Figure 1 shows the overall architecture of YOLOv5.

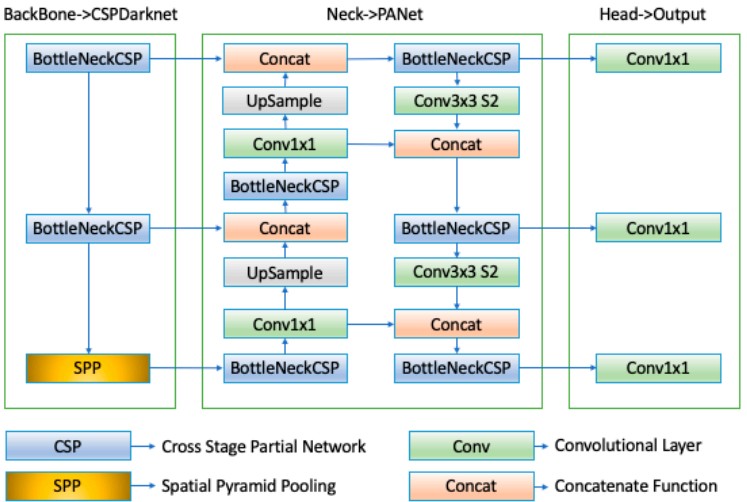

**Figure 1.** The YOLOv5 architecture.

The location of the bounding box is calculated using Equation (1):

$$\cup_x^y = \mathcal{P}_{x,y} * IOU_{predicted}^{ground\ truth} \tag{1}$$

In Equation (1), $x$ and $y$ are the $y_{th}$ bounding box of the $x_{th}$ grid. $\cup_x^y$ is the probability score for the $y_{th}$ bounding box of the $x_{th}$ grid. $\mathcal{P}_{x,y}$ equals 1 when there is a target and 0 when there is no target in the $y_{th}$ bounding box. The IoU $IOU_{predicted}^{ground\ truth}$ is the IoU between the ground truth and the predicted class. Higher IoUs mean more accurately predicted bounding boxes.

The loss function of YOLOv5 is the combination of loss functions for the bounding box, classification, and confidence. Equation (2) represents the overall loss function of YOLOv5 [32]:

$$loss_{YOLOv5} = loss_{bounding\ box} + loss_{classification} + loss_{confidence} \tag{2}$$

$loss_{bounding\ box}$ is calculated using Equation (3):

$$loss_{bounding\ box} = \lambda_{if} \sum_{a=0}^{b^2} \sum_{c=0}^{d} E_{a,c}^g h_g (2 - K_a X n_a) \left[ \left( x_a - x'^c_a \right)^2 + \left( y_a - y'^c_a \right)^2 + \left( w_a - w'^c_a \right)^2 + \left( h_a - h'^c_a \right)^2 \right] \tag{3}$$

In Equation (3), the width and height of the target object are denoted using $h\prime$ and $w\prime$. $x_a$ and $y_a$ indicate the coordinates of the target object in an image. Finally, the indicator function ($\lambda_{if}$) shows whether the bounding box contains the target object.

$loss_{classification}$ is calculated using Equation (4):

$$loss_{classification} = \lambda_{\text{classification}} \sum_{a=0}^{b^2} \sum_{c=0}^{d} E_{a,c}^{g} \sum_{C \in c_l} L_a(c) \log(LL_a(c)) \tag{4}$$

$loss_{confidence}$ is calculated using Equation (5):

$$loss_{confidence} = \lambda_{\text{confidence}} \sum_{a=0}^{b^2} \sum_{c=0}^{d} E_{a,c}^{confidence} (c_i - c_l)^2 + \lambda_g \sum_{a=0}^{b^2} \sum_{c=0}^{d} E_{a,c}^{g} (c_i - c_l)^2 \tag{5}$$

In Equations (4) and (5), $\lambda_{\text{confidence}}$ indicates the category loss coefficient, $\lambda_{\text{classification}}$ the classification loss coefficient, $c_l$ the class, and c the confidence score.

*Construction of the Experiment and Data Acquisition*

We collected drone and bird images from public resources such as Google, Kaggle, Flicker, Instagram, etc. The drone images came from different altitudes, angles, backgrounds, and views, ensuring variability in the dataset. The bird images consisted of 300 different species. The entire dataset was formed using 479 bird images and 1916 drone images; altogether, the dataset consisted of 2395 images. We used a 70:30 train:test split to train and test the YOLOv5 model. The training dataset had 1677 images and the testing dataset had 718 images. We used data augmentation techniques to overcome data scarcity. In fact, using 3 variants of data augmentation, we generated a total of 5749 images. Using a freely available labeling tool, we annotated the images and divided them into two classes. Drone images were annotated as "first class" and bird images as "zero class." YOLO implementation requires that all images be saved in the. txt format, which has four coordinates for the object, including the class of 0 or 1.

We collected two videos of drones flying, using our own two drones: a DJI Mavic Pro and DJI Phantom III. We captured video shots at three different altitudes: 60 feet, 40 feet, and 20 feet. These are altitudes commonly used by drone pilots, especially drone hobbyists. At 60 feet, the drone looked almost like a bird. We captured the videos to evaluate the performance of the YOLOv5 model in terms of accuracy and speed, mainly at high altitudes.

We conducted the experiment using Google CoLab, a free cloud notebook in which we wrote the code, to implement YOLOv5. We fine-tuned the original YOLOv5 to train and test the model using our customized dataset. To accelerate and improve detection accuracy, we used a transfer learning technique. We employed the weights that were already available with the original YOLOv5 to implement transfer learning. We trained our customized model on top of the YOLOv5s.pt weight that was saved while training YOLOv5 on the COCO dataset. The original YOLOv5 was implemented using PyTorch. We also chose PyTorch. At the time we trained our model, Google CoLab allocated a Tesla T4 with a 15110MiB memory NVIDIA GPU. To fine-tune YOLOv5, we chose the values of the various hyperparameters suggested in the original. We used an lr of 0.01, momentum of 0.937, and decay of 0.0005. The model was optimized using stochastic gradient descent. The albumentations were Blur ($p = 0.01$, blur_limit = (3,7)), MedianBlur ($p = 0.01$, blur_limit = (3,7)), ToGray ($p = 0.01$), and CLAHE ($p = 0.01$, clip_limit = (1,4.0) title_grid_size = (8,8)).

We changed the number of classes from 80 to 2, since we had 2 classes: drone and bird. The model had 214 layers with 7,025,023 parameters, 7,025,023 gradients, and 16.0 GFLOPs. We used the Roboflow API to load and perform the data augmentation and preprocessing. We used the same dataset employed in our previous experiment. We performed auto-orient and modified classes as data preprocessing techniques. Auto-orient is an image processing technique that ensures that images match the source device orientation. Sometimes, the coordinates in various cameras may confuse (x,y) and (y,x). Auto-orient prevents bad data from being fed into YOLOv5. In addition to data preprocessing, we performed data augmentation using Roboflow API, which helped reduce the data shortage issue. We set the parameters as follows: flip: horizontal, hue: between −25 degrees and +25 degrees, cutout:

3 boxes with 10% size each, and mosaic: 1. Data augmentation ensured data variability, artificially generating 5749 total images, and randomly splitting the entire dataset into a 70:30 train:test split. Figure 2 shows the augmented dataset. We trained the model for 4000 iterations and saved the best weight to test the model using the testing images and videos. During training, we used %tensorboard to log the runs that autogenerated the learning curves in order to evaluate the model's performance beyond the evaluation metrics. Figure 3 shows a flowchart of the overall conducted experiment.

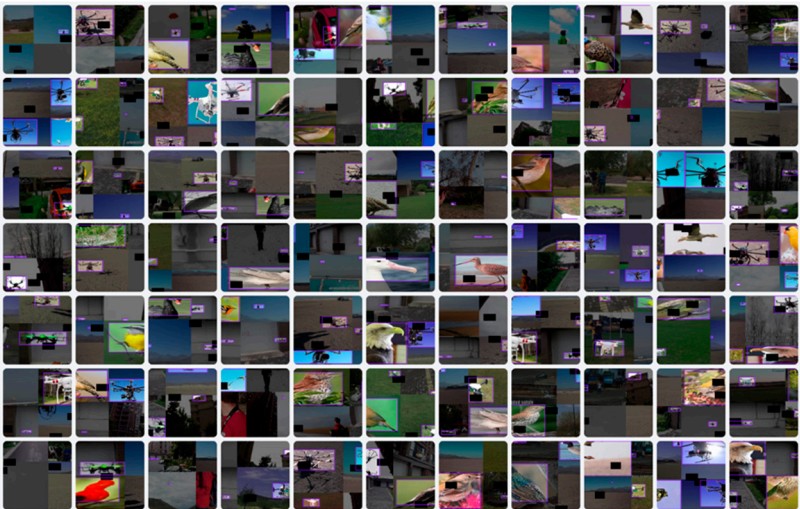

**Figure 2.** Augmented dataset.

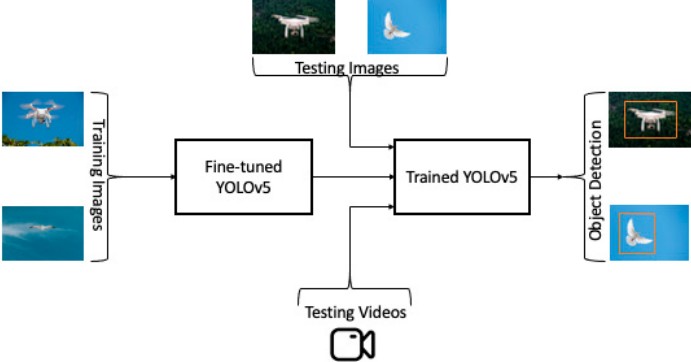

**Figure 3.** Overall conducted experiment flowchart.

## 4. Results

We evaluated the trained model using the mAP, precision, recall, and F1-scores. We used FPS as the evaluation metric to evaluate the speed of detection in the videos. Table 1 shows the mAP, precision, recall, and F1-scores. The model was evaluated on a testing dataset from a random train:test split. The testing images had data variabilities in terms of different backgrounds (e.g., bright, dark, blur, etc.) and weather conditions (e.g., cloudy, sunny, foggy, etc.), as well as images with multiple classes. To track the evaluation metrics, we plotted the values across iterations. Figure 4 shows the overall training summary of the model. The loss curves indicate a downward trend, meaning that during training, the losses were minimized both for training and validation. The metrics curves show upward trends, meaning the performance of the model improved over the iterations during training. We plotted the precision-recall curve to evaluate the model's prediction preciseness (see Figure 5). The curve tended towards the right top corner, meaning that the values were mostly close to one (i.e., the rate of misclassification was very low when using this model).

**Table 1.** Overall and individual evaluation metrics results.

| Class | Precision | Recall | mAP50 |
|---|---|---|---|
| All | 0.918 | 0.875 | 0.904 |
| Bird | 0.860 | 0.766 | 0.820 |
| Drone | 0.975 | 0.985 | 0.987 |

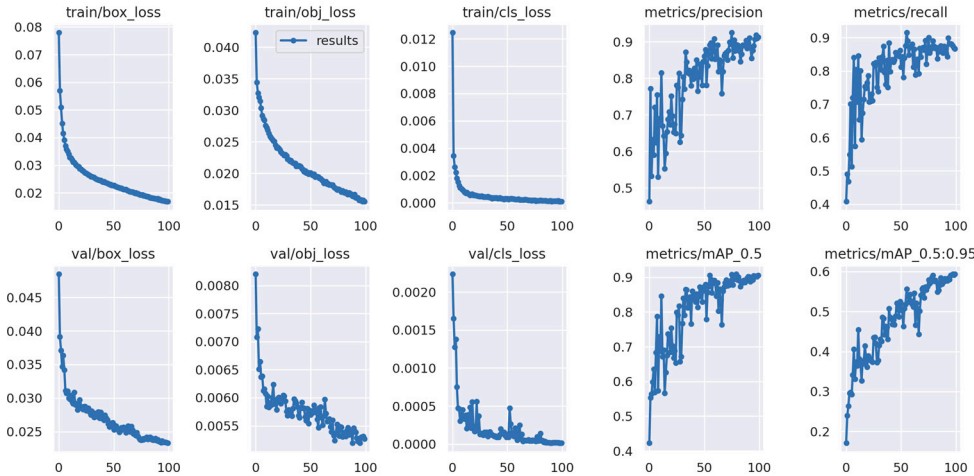

**Figure 4.** Overall summary of training.

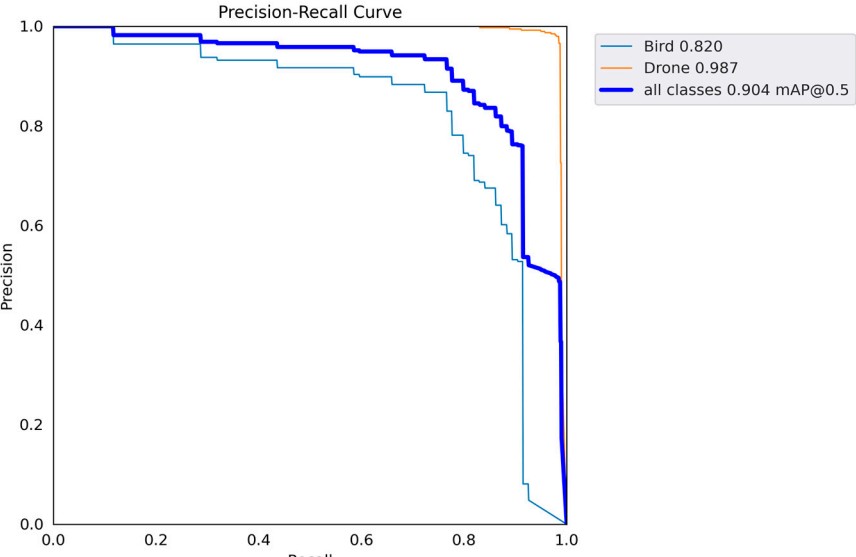

**Figure 5.** Precision-recall curve.

Finally, we show the model's evaluation metrics in Table 1, which offers an overall summary of the results regarding the trained model's performance when using the testing images. We achieved precision, recall, and mAP50 values of 0.918, 0.873, and 0.904 for all images, respectively. In addition, we calculated individual precision, recall, and mAP50 values for Classes 1 and 2 (see Table 1). Figure 6 shows the drones predicted by the model using randomly chosen testing images. Figures 7–12 show the predictions with bounding boxes and class scores at three different altitudes (20 ft, 40 ft, and 60 ft) in videos using two different types of drones (the DJI Mavic Pro and DJI Phantom III, Da-Jiang Innovations, Shenzhen, China).

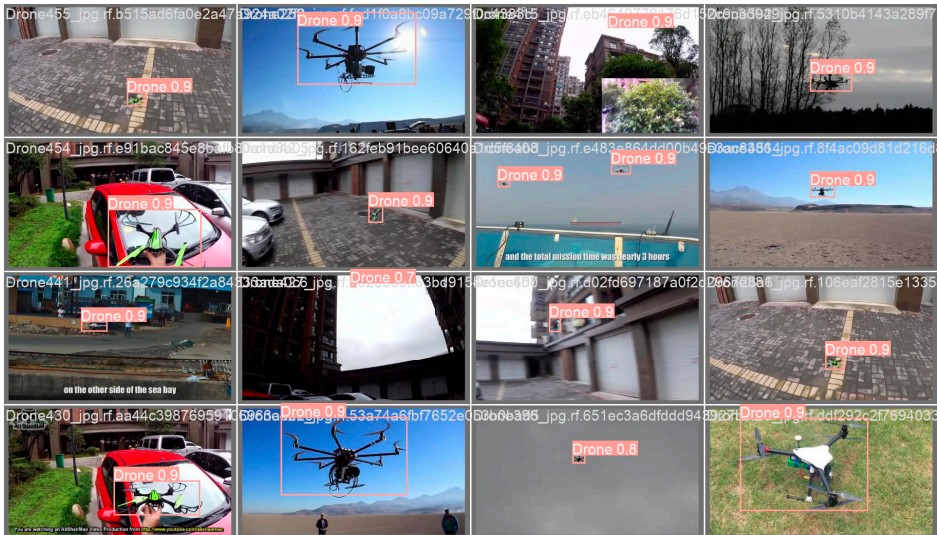

**Figure 6.** Drone predictions for test images.

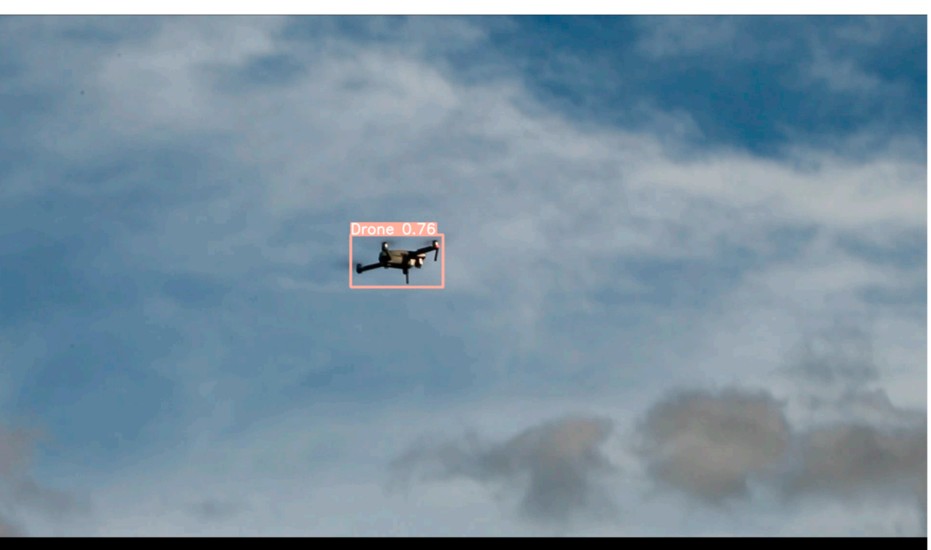

**Figure 7.** At 20ft, DJI Mavic Pro.

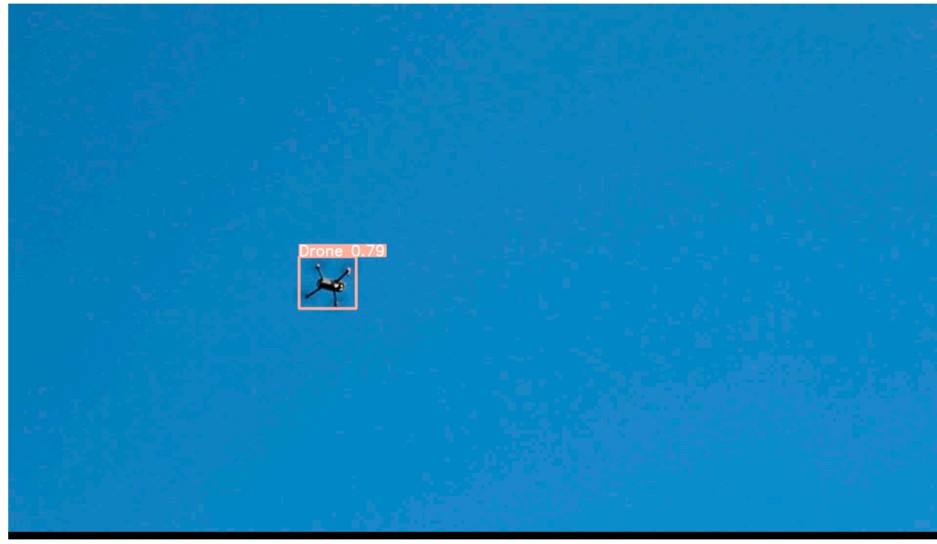

**Figure 8.** At 40ft, DJI Mavic Pro.

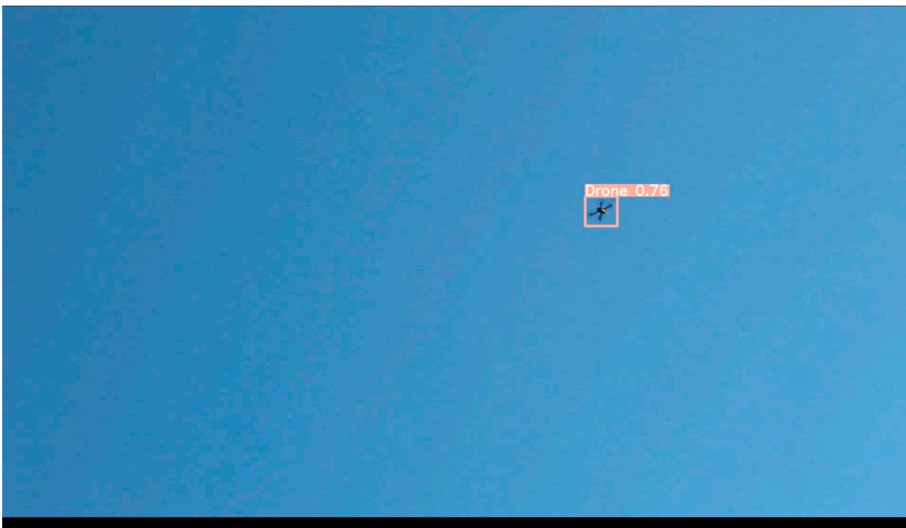

**Figure 9.** At 60ft, DJI Mavic Pro.

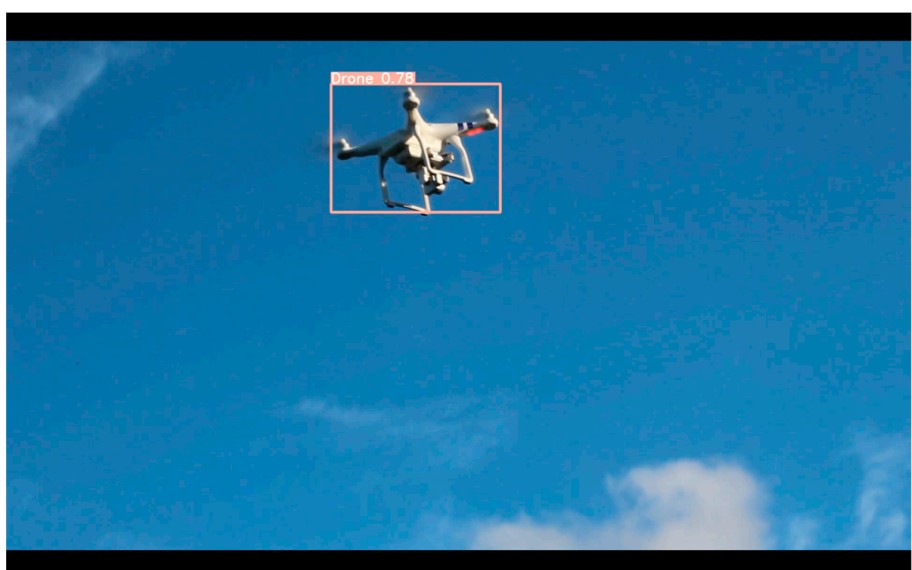

**Figure 10.** At 20ft, DJI Phantom III.

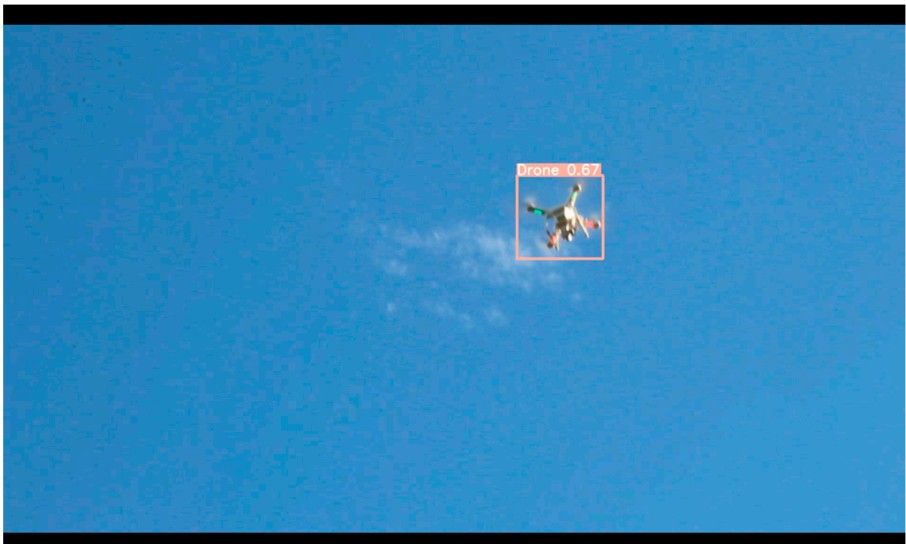

**Figure 11.** At 40ft, DJI Phantom III.

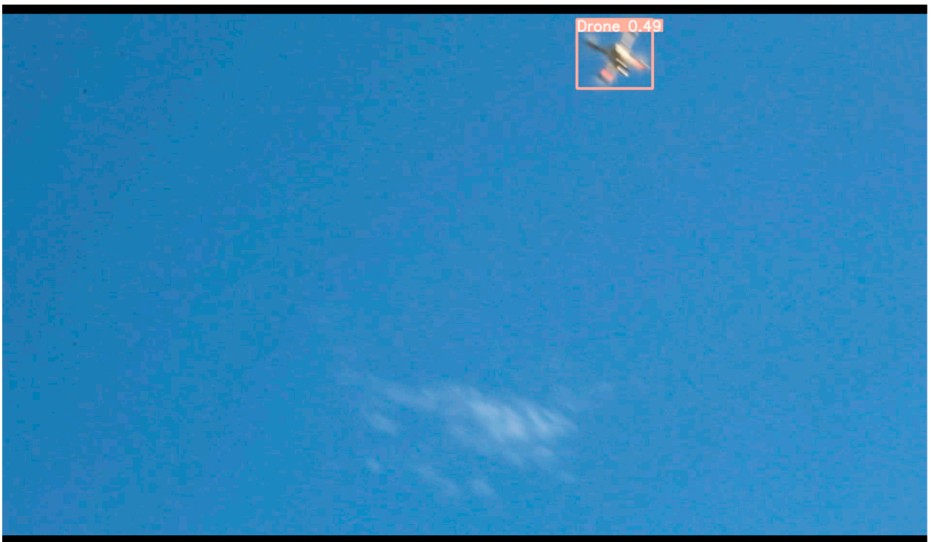

**Figure 12.** At 60 ft, DJI Phantom III.

Appendix A contains more predictions based on the trained YOLOv5 model. While the model worked well on the majority of the test images, there were a few instances of misclassification. Figures A7 and A8 show two misclassifications in which the model misidentified certain drone-like objects as drones alongside correct predictions in these images. Blurred photos might be one of the causes of such misclassification. We can address this problem by employing more training photos, which is outside the scope of this study. The prediction confidence scores were poor, hovering around 10%. We might perhaps establish a confidence score threshold to avoid such misclassification while increasing the number of training images. There were just a few "bird" classes. Figure A3 depicts an example of correct "drone" and "bird" predictions. However, because of the uncertainty of both classes in a single video frame, we were unable to do any drone and bird detection in videos.

## 5. Model Complexity and Parameter Uncertainty

To do a quicker prediction, we employed YOLOv5, which mainly relies on GPU implementation. GPU implementation complicates CPU deployments. Data augmentation techniques such as rotation and flipping were used to artificially supplement the dataset for improved training and performance. The parameter uncertainty in our experiment included sampling errors, overfitting, and so forth. Too many classes from one class may create sampling error, whereas training a smaller number of images with higher parameters may result in overfitting. We used pre-trained model weights that were trained on the COCO dataset, and we trained our fine-tuned model on top of the pre-trained weights.

## 6. Discussion

Using deep learning for the detection of drones has become a common topic in the research community, due to the substantial importance of restricting drones in unauthorized regions; however, improvement is still needed. The authors of [30] proposed a drone detection methodology using deep learning, employing YOLOv3 to detect and classify drones. More than 10,000 different categories of drones were used to train the algorithm, and a mAP of 0.74 was achieved at the 150th epoch. Though they used a YOLO-based approach, their study did not consider testing the model using videos, different weather conditions, and backgrounds; most importantly, they did not test their model using images of objects like drones. The authors of [33] used deep learning-based techniques and Faster R-CNN on a dataset created from videos collected by the researchers. The following image augmentation techniques were employed: geometric transformation, illumination variation, and image quality. The researchers did not calculate the mAP values and instead plotted a

precision-recall (AUC) curve to evaluate the performance. Using a synthetic dataset, their model achieved an overall AUC score of 0.93; for a real-world dataset, their model achieved an overall AUC score of 0.58. The dataset was trimmed from video sequences, and thus had no objects much of the time. In our previous research, we analyzed the performance of our proposed methodology using YOLOv4 and showed that the proposed methodology outperformed existing methodologies in terms of mAP, precision, recall, and F-1 scores. Using YOLOv4, we were able to achieve a mAP of 0.7436, precision of 0.95, and recall of 0.68. Most importantly, we included another evaluation metric, FPS, to evaluate the performance, achieving an average FPS of 20.5 for the DJI Phantom III videos and 19.0 FPS for the DJI Mavic Pro videos, all at three different high altitudes (i.e., 20 ft, 40 ft, and 60 ft). We tested the model using a highly variable dataset with different backgrounds (e.g., sunny, cloudy, dark, etc.), various drone angles (e.g., side view, top view, etc.), long-range drone images, and multiple objects in a single image. Our previous methodology achieved such an improvement due to the real-time detection capability of YOLOv4 acting as a single-stage detection process, and the various new features of YOLOv4 (e.g., CSP, CmBN, mish activation, etc.), which sped up detection. Furthermore, the default MOSAIC = 1 flag automatically performed the data augmentation. In this research, we employed Google CoLab and Google Deep Learning VM for parts of the training and testing. In addition to YOLOv4, YOLOv5 showed performance improvement, as shown in [1]. They obtained a precision of 0.9470, a recall of 0.9250, and a mAP of 0.9410. Although their evaluation metrics were higher than ours, our dataset was bigger. Furthermore, we had binary classes, whereas they just had a "drone" class. They did not employ data augmentation, whereas we used a data augmentation technique to build a collection of over 5700 images. As a result of the variability in the dataset and the addition of new classes with data augmentation, our suggested technique is resilient and scalable in real-world scenarios.

Our results for the present research outperformed our previous methodology, achieving a mAP of 0.904. Because of the lightweight design, YOLOv5 recognized objects faster than YOLOv4. YOLOv4 was created using darknet architecture; however, YOLOv5 is built with a PyTorch framework rather than a darknet framework. This is one of the reasons we obtained more accuracy and speed than earlier methodologies. In addition to the architecture itself, we fine-tuned the last layers of the original YOLOv5 architecture so that it performed better on our customized dataset. Other than the layer tuning, we customized the default values of the learning rate, momentum, batch size, etc. We trained the model for 100 iterations since we trained the custom dataset on top of the transferred weights for the COCO dataset. In addition to mAP, we achieved a precision of 0.918, recall of 0.875, and F-1 score of 0.896. In terms of F-1 score and recall, we also outperformed the previous model. We further tested the new model on two videos, using a Tesla T4 GPU. For the DJI Mavic Pro, we achieved a maximum FPS of 23.9 ms, and for the DJI Phantom III, a maximum FPS of 31.2 ms. Thus, in terms of inference speed, we also outperformed the previous model's performance. We achieved this improvement due to the new feature additions included in YOLOv5, such as the CSPDarknet53 backbone, which resolved the gradient issue using fewer parameters, and thus was more lightweight. Other helpful feature additions included the fine-tuning of YOLOv5 for our custom dataset, data augmentation performed to artificially increase the number of images, and data preprocessing to make training the model smoother and faster. The evaluation metric, F1 score, is a weighted sum of precision and recall. Precision is the accuracy of positive class prediction, whereas recall is the proportion of true positive classes. The greater the F1 score, the better the model in general. The correctness of bounding boxes in objects is measured by mAP, and the greater the value, the better. The speed of object detection is measured in frames per second (FPS). Table 2 compares the performance of the previous and proposed models in terms of four evaluation metrics: precision, recall, F1 score, mAP50, and FPS.

**Table 2.** Comparison between previous and proposed models' performance.

| Models | Precision | Recall | F1 Score | mAP50 | | FPS |
|---|---|---|---|---|---|---|
| Previous YOLOv4 [30] | 0.950 | 0.680 | 0.790 | 0.7436 | DJI Mavic Pro | 19.0 ms |
| | | | | | DJI Phantom III | 20.5 ms |
| Proposed YOLOv5 | 0.918 | **0.875** | **0.896** | **0.9040** | DJI Mavic Pro | **23.9 ms** |
| | | | | | DJI Phantom III | **31.2 ms** |

## 7. Conclusions

In this research, we compared the performance of one of the latest versions of YOLO, YOLOv5, to our previously proposed drone detection methodology that used YOLOv4. To make a fair comparison, we employed the same dataset and the same computing configurations (e.g., GPU). We first fine-tuned the original YOLOv5, as per our customized dataset that had two classes: bird and drone. We further tuned the values of the hyperparameters (e.g., learning rate, momentum, and decay) to improve the detection accuracy. In order to speed up the training, we used transfer learning, implementing the pre-trained weights provided with the original YOLOv5. The weights were trained on a popular and commonly used dataset called MS COCO. To address data scarcity and overfitting issues, we used data augmentation via Roboflow API and included data preprocessing techniques to smoothly train the model. To evaluate the model's performance, we calculated the evaluation metrics on a testing dataset. We used precision, recall, F-1 score, and mAP, achieving 0.918, 0.875, 0.896, and 0.904 values, respectively. We outperformed the previous model's performance by achieving higher recall, F-1 score, and mAP values (a 21.57% improvement in mAP). Furthermore, we tested the speed of detection on videos of two different drone models, the DJI Phantom III and the DJI Mavic Pro. We achieved maximum FPS values of 23.9 and 31.2, respectively, using an NVIDIA Tesla T4 GPU. The videos were taken at three altitudes—20 ft, 40 ft, and 60ft—to test the capability of the detector for objects at high altitudes. In future work, we will use different versions of YOLO and larger datasets. In addition, other algorithms for object detection will be included to compare the performance. Various drone-like objects such as airplanes will be added as classes alongside birds to improve the model's ability to distinguish among similar objects.

**Author Contributions:** Conceptualization, S.S. and B.A.; methodology, B.A. and S.S.; software, S.S.; validation, B.A. and S.S.; formal analysis, S.S and B.A.; investigation, B.A. and S.S.; resources, S.S.; data curation, S.S.; writing—original draft preparation, S.S.; writing—review and editing, B.A.; visualization, S.S.; supervision, B.A.; project administration, S.S. and B.A.; funding acquisition, B.A. All authors have read and agreed to the published version of the manuscript.

**Funding:** This research received no external funding.

**Institutional Review Board Statement:** Not applicable.

**Informed Consent Statement:** Not applicable.

**Data Availability Statement:** The datasets used or analyzed during the current study are available from the corresponding author upon reasonable request.

**Conflicts of Interest:** We declare that there is no conflict of interest.

## Abbreviations

| | |
|---|---|
| AUC | Area under the ROC Curve |
| CCTV | Closed-Circuit Television |
| CLAHE | Contrast Limited Adaptive Histogram Equalization |
| CNN | Convolutional Neural Network |
| COCO | Common Objects in Context |
| CPU | Central Processing Unit |
| CSPNet | Cross-Stage Partial Network |

| DJI | Da-Jiang Innovations |
| FLOPS | Floating-Point Operations per Second |
| FPS | Frames per Second |
| GFLOP | Giga Floating Point Operations per Second |
| Google CoLab | Google Colaboratory |
| GPU | Graphical Processing Unit |
| IoU | Intersection over Union |
| KNN | k-nearest neighbor |
| lr | Learning Rate |
| mAP | Mean Average Precision |
| MATLAB | Matrix Laboratory |
| PANet | Path Aggregation Network |
| R-CNN | Region-Based Convolutional Neural Network |
| ResNet | Residual Network |
| SSD | Single-Shot Multi-box Detector |
| SVM | Support Vector Machine |
| UAV | Unmanned Aerial Vehicle |
| VM | Virtual Machine |
| YOLO | You Only Look Once |
| YOLOv2 | You Only Look Once version 2 |
| YOLOv3 | You Only Look Once version 3 |
| YOLOv4 | You Only Look Once version 4 |
| YOLOv5 | You Only Look Once version 5 |

## Appendix A

In a variety of photos, our classifier effectively identified drone and bird objects. We evaluated images with intricate backgrounds and various climatic conditions. Here we have the detection results, where the images are displayed together with their corresponding class names and class probabilities. YOLOv5 generated the predictions in batches. Thus, predictions are shown all in one figure. Additionally, we tested images that have "drone" and "bird" in one image. In augmented training images, 0 refers to "bird" and 1 refers to "drone".

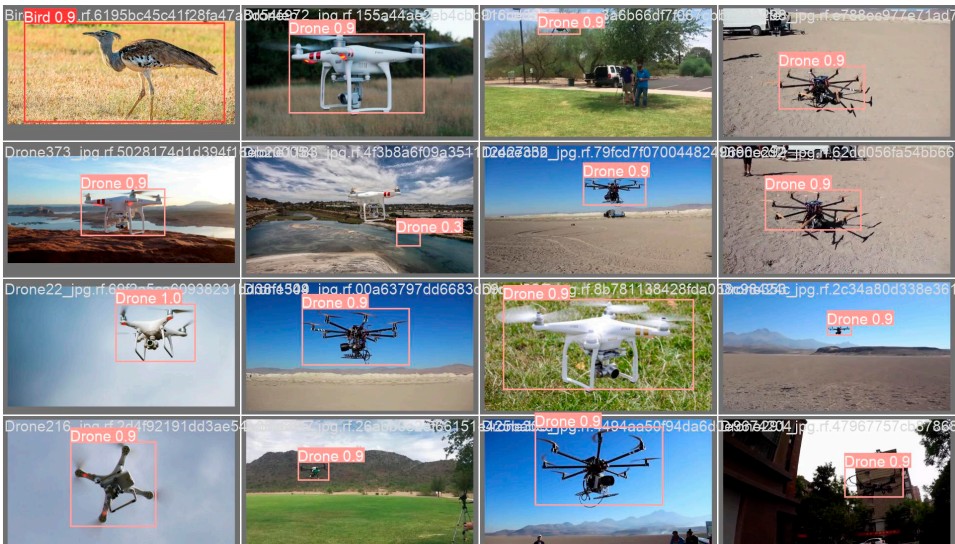

**Figure A1.** First batch prediction by YOLOv5.

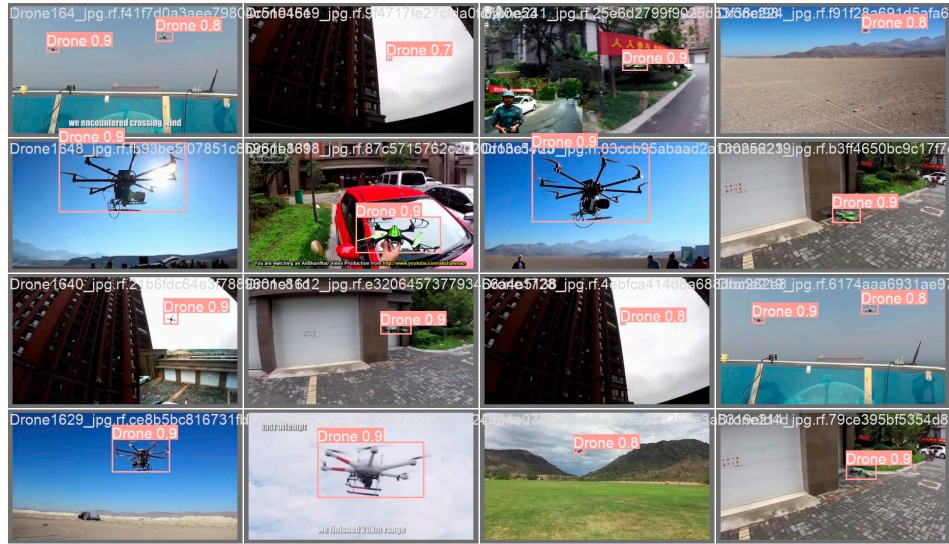

**Figure A2.** Second batch prediction by YOLOv5.

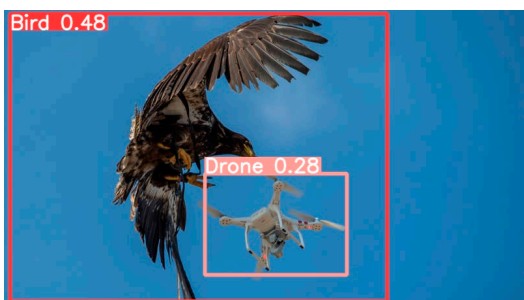

**Figure A3.** Bird and drone in images predicted by YOLOv5.

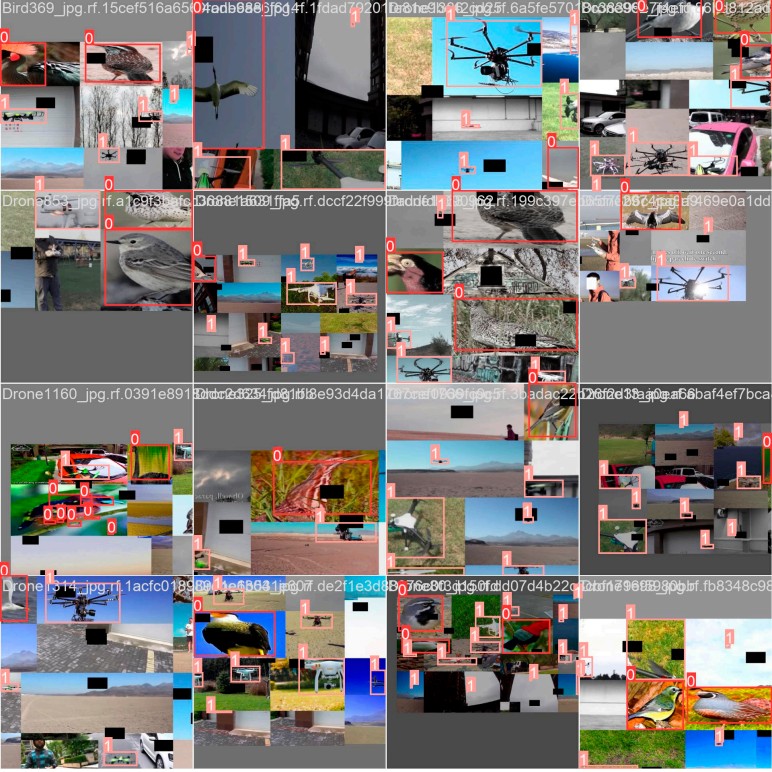

**Figure A4.** First batch of augmented training image predicted by YOLOv5.

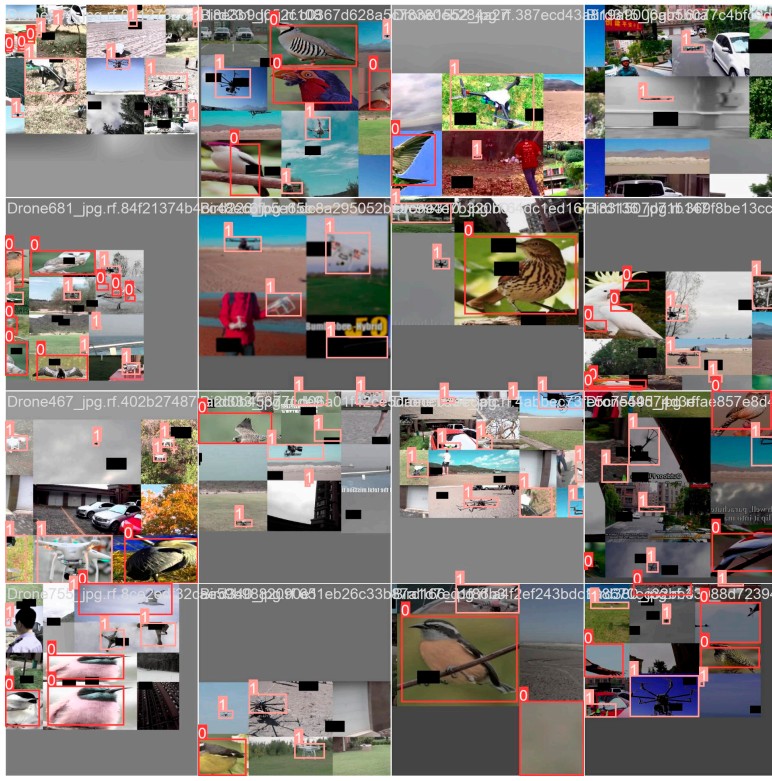

**Figure A5.** Second batch of augmented training image predicted by YOLOv5.

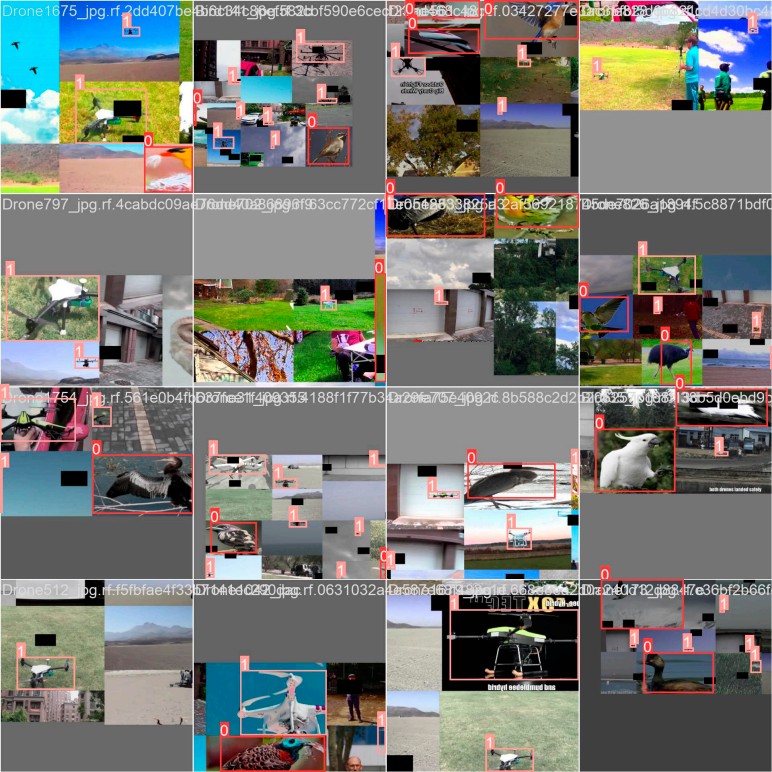

**Figure A6.** Third batch of augmented training image predicted by YOLOv5.

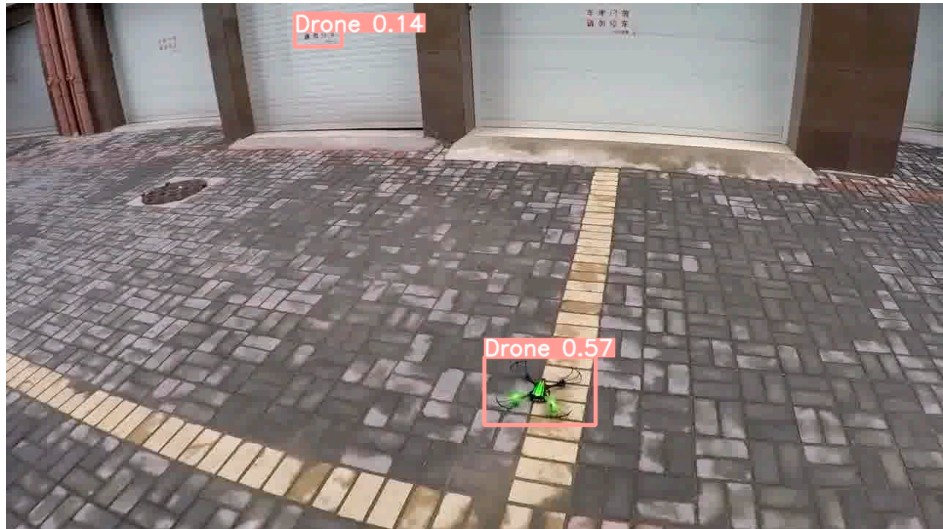

**Figure A7.** Instance1 of misclassified image predicted by YOLOv5.

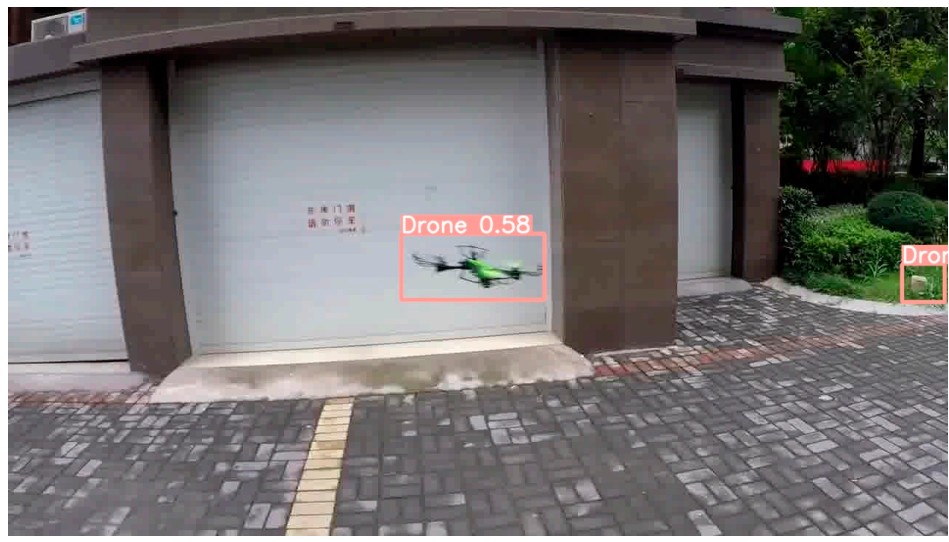

**Figure A8.** Instance2 of misclassified image predicted by YOLOv5.

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
