# Peer review of "Drone Detection Using YOLOv5"

_2673-4117, doi:10.3390/eng4010025_

Round 1
Reviewer 1 Report
§ This paper deals with an exciting topic Drone Detection using YOLOv5. All the acronyms should be defined and explained first before using them such that they become evident for the readers.
§ The Introduction and related work parts give valuable information for the readers as well as researchers. In addition, recent papers should be added in the part of related work.
§ Representation of figures needs to be improved.
§ The importance of the design carried out in this manuscript can be explained better than other important studies published in this field. I recommend the authors to review other recently developed works. I will recommend following manuscript to be referred.
- Al-Qubaydhi, Nader, Abdulrahman Alenezi, Turki Alanazi, Abdulrahman Senyor, Naif Alanezi, Bandar Alotaibi, Munif Alotaibi, Abdul Razaque, Abdelaziz A. Abdelhamid, and Aziz Alotaibi. "Detection of Unauthorized Unmanned Aerial Vehicles Using YOLOv5 and Transfer Learning." Electronics 11, no. 17 (2022): 2669.
· What makes the proposed method suitable for this unique task? What new development to the proposed method have the authors added (compared to the existing approaches)? These points should be clarified.
§ Discussion” section should be added in a more highlighting, and argumentative way. The author should analysis the reason why the tested results is achieved.
§ The authors should clearly emphasize the contribution of the study. Please note that the up-to-date of references will contribute to the up-to-date of your manuscript. Also, indicate the contribution in the “Introduction” section.
§ The complexity of the proposed model and the model parameter uncertainty are not enough mentioned.
§ It will be helpful to the readers if some discussions about insight of the main results are added.
Author Response
Thank you so much for your feedback. Please see the attached word document.

Reviewer 2 Report
1. Improve Introduction. The topics discussed should flow (background, motivation, statement of research, objectives, etc)
2. Line 59-60: "there are currently three state-of-the-art object detection techniques available" is inaccurate. Authors should reevaluate their statements.
3. The paragraph that runs through lines 57 - 77 contains several inaccurate assumptions about object detectors. The authors can discuss a few object detectors and not generalise the properties of all object detectors.
4. Lines 57 - 77 should be reevaluated entirely and rewritten.
5. Separate Introduction and Literature Review (another section).
6. Figure 3 does not present the experiment setup or the image does not accurately portray the experiment setup.
7. Section 3.1 can be part of Materials and Methods.
8. Can the dataset split (90:10) be increased to (70:30). The authors can also consider k-fold cross-validation.
9. Can we have a table that compares your result with your previous work and maybe some other papers you have mentioned here?
10. Was there any case of birds in the video used to test? Can we have some illustrations of this scenario in the discussion section?
Thank you.
Author Response
Thank you very much for your feedback. Please see the attached document.

Reviewer 3 Report
This paper presents a method based on Yolo for drone image detection. It is well-written. Here are some issues need to improve.
- The dataset is quite small. However the ration train:test (90:10) is not reasonable. Just 235 images for testing and the high accuracy (or F1) is achieved. We suggested to change the ratio or collect more data for testing set.
- In the Results section, could you show some failed cases and discuss why the model failed in these cases ?
- Some related references should be cited:
https://doi.org/10.1155/2021/7918165
https://doi.org/10.1155/2021/7748350
https://doi.org/10.1155/2022/8924027
Author Response

(The authors gave the same response as above.)

Reviewer 4 Report
This paper is entitled “Drone Detection using YOLOv5."
This paper provides us with a method to detect Drone(s) using one of the state-of-the-art object detection methods (YOLOv5). The author(s) used YOLOv5 and the Transfer learning method in this research. The paper is aimed at a topic that is very modern and interesting nowadays. The Artificial Intelligence (AI) and Computer Vision (CV) systems and solutions will provide independence and autonomy to the robotic systems regardless if they are in the air, on the ground, or in the sea. For this reason, I strongly encourage you to continue with your research in this scientific domain and to keep exploring the possibilities provided by AI and CV.
Although your manuscript falls within the aim and scope of this journal, and the problems being addressed are potential of interest to our readership, it is being declined due to a lack of sufficient novelty. I can hardly see the main contribution of this manuscript because the manuscript focuses on explaining how to enable YOLO as an object detection method using some new datasets from Google, Kaggle, Flicker, Instagram, etc. Also, It seems that there is no new deep learning architecture, no intriguing insight about enabling deep learning, no theoretical analysis on deep learning algorithms, and no empirical innovations on empirical test-bed or systems. And I think this manuscript does not meet the required quality standards to be considered for research publication. More intensive work on learning architecture and embedding the architecture is needed to substantiate the conclusions in your manuscript.
My comments about this paper are as below:
1. In the abstract, the authors explain the whole research, using original YOLOv5 to detect Drone(s) and employ pre-trained weights and data augmentation.
2. Introduction and Background of this research are well explained.
3. Representation of figures is Inadequate for figures 7-12 (too big)
4. Why only two classes? In the real world, you must consider many flying items (such as airplanes, ballon, kites, etc.). Because the author used COCO weight which has many classes.
Table 1:
> it would be nicer to have the best values highlighted.
Overall, this manuscript is not qualified to be accepted for publication in this journal.
Author Response

(The authors gave the same response as above.)

Round 2
Reviewer 1 Report
The authors partially responded to my first two comments, but the rest of the comments have not been responded properly. So I still want the authors to work on previous comments and provide complete justification.
Reviewer 3 Report
The authors have been revised carefully our comments. I think it is ready for publication.
Author Response
Thank you for your valuable feedback that made our paper better.
Reviewer 4 Report
For me, fine-tuning the original algorithm based on a customized dataset does not have significance in terms of academic research contribution. Besides that, the writing style and paper representation are adequate for publication.
Author Response
Thank you again for your valuable feedback. We appreciate your point of view regarding fine-tuning. We feel that this study adds significant value to the research domain of detecting drones in unauthorized/restricted areas. The results of this study might also help public safety organizations.